# Working with Waste: Hazards and Mitigation Strategies Used by Waste Pickers in the Inner City of Durban

**DOI:** 10.3390/ijerph192012986

**Published:** 2022-10-11

**Authors:** Ntobeko Mlotshwa, Tanya Dayaram, Asiphile Khanyile, Princess A. Sibanda, Kira Erwin, Tamlynn Fleetwood

**Affiliations:** 1Urban Futures Centre, Faculty of Engneering and Built Environment, Durban University of Technology, Steve Biko Campus, Durban 4001, South Africa; 2Groundwork, 8 Gough Road, Pietermaritzburg 3201, South Africa; 3NRF SARChI Chair in Sexualities, Genders and Queer Studies, University of Fort Hare, Alice 5700, South Africa

**Keywords:** waste management, recycling, occupational hazard, risk mitigation, waste pickers, South Africa

## Abstract

Informal waste pickers in cities across the Global South divert significant amounts of tonnage from landfills. This diversion contributes towards a sustainable environment and better public health practices. Informal workers globally derive livelihoods from collecting, sorting, and selling recyclable waste. In South Africa, there is growing recognition of the valuable work that waste pickers carry out. Despite this, however, these informal workers remain largely unrecognised, are often stigmatised, and suffer from a lack of social protection linked to their work. This lack of recognition and protection creates specific occupational hazards for waste pickers. Using an ethnographic method, this study explores the physical and socio-psychological hazards that emerge from waste picking on the streets of the inner city of Durban, in South Africa. We found that the waste pickers, the majority of whom were women, developed mitigation strategies against these risks. A better understanding of how the occupational hazards of waste picking are shaped by the local context of working on the street enables the recognition of the knowledge waste pickers already hold regarding mitigation strategies. Insight into occupational hazards are important to consider if the municipal integration of waste pickers is to happen in a way that ensures access to social protections for these informal workers.

## 1. Introduction

Rapid urbanisation globally, but significantly in Africa [1], has driven a subsequent increase in waste generation linked to urban consumer lifestyles. Under current production and consumption patterns, the World Bank estimates an increase in waste generation of “73% from 2020 levels to 3.88 billion tonnes in 2050” [2]. The management of this waste is one of the key challenges facing both developing and developed countries today [3]. Many cities in the Global South suffer from overburdened waste management systems, limited resources for implementing more effective management systems, and a lack of landfill spaces [4]. A lack of effective and efficient waste management leads to uncollected waste, illegal dumping, and open burning. Both official waste management systems and unofficial systems, such as open burning and illegal dumping, have profound negative impacts on climate change (notably via the release of GHG and methane from poorly regulated landfills, which drives global warming) and human health and wellbeing (mostly through the ingestion of toxins from air, soil, and water polluted by waste). Poorly managed waste also has a negative impact on urban infrastructure, such as the blockage of drains and sewage pipes, which in turn exacerbates adverse weather events caused by climate change such as urban flooding. Environmental activists have long fought for a shift in regulating the production of waste products. The United Nations Plastics Treaty and Extended Producer Responsibility Acts aim to push producers into taking responsibility for their waste products to reduce (at worse) and eliminate (at best) forms of packaging that cannot be recycled or reused, such as single-use plastics. On a macroscale, poor waste management is a direct threat to both public health and the planetary environment [5]. The growing awareness of this threat has seen a significant shift globally in recycling certain waste streams, particularly paper, cardboard, tin, and some plastics. 

Working with waste at a microscale, however, has notable occupational health and social-ecological impacts. This article focuses on some of the occupational hazards for waste pickers, the people who work the most intimately, and overwhelmingly informally, with waste in cities in the Global South. Waste pickers in South Africa are perceived as informal workers who receive no social protections, and as such are an important sector for investigating the occupational hazards associated with vulnerable and precarious work. 

This article focuses on the occupational risks that face waste pickers working in the inner city of Durban, situated within the eThekwini Municipality on the east coast of South Africa. The eThekwini Municipality has an area of 2297 km² and incorporates large tracts of urban, rural, and peri-urban land. With reference to the 2011 Census data, the eThekwini Municipality had a population of approximately 3.5 million people, with just over 25% of the population under 15 years of age. Disaggregated by race, in 2011, 73.8% of the population was Black African, 16.7% were classified as Indian/Asian, the White population was 6.6%, and the Coloured population was in the minority (3%) [6]. The eThekwini Municipality contributes 59,88% to the KwaZulu-Natal gross domestic product (GDP) and 9.59% to the national GDP. In 2018, the main economic sectors in the municipality included finance and community services, which were the largest sectors at 21% each; manufacturing (19%); trade (17%); and transport (14%) [7]. Whilst the formal sectors of the economy continue to provide the most job opportunities in the city, in 2018, 26% of employment in Durban was informal. Of this 26%, informal wage employment constituted the biggest share (17%), with 9% of individuals reporting to be self-employed informally [8]. Waste-pickers are self-employed informal workers, although data on their exact numbers and economic contribution are lacking. Reported estimates of waste-picker numbers in South Africa vary from between sixty and ninety thousand to as high as two hundred and fifteen thousand [9]. 

This article offers some initial context for the occupational hazards of waste pickers globally and in South Africa, as well as the critical environmental and public health benefits these informal workers provide across cities. It draws on narrative data from an ethnographic study with eight waste pickers in the inner-city area to explore what occupational risks emerge from their work. Some of these, such as the physical risks, mirror the hazards experienced by waste pickers working on landfills. Others, such as the socio-psychological and environmental hazards, resonate with research that explores the health risks of waste pickers elsewhere, but also take more specific forms shaped by the context of working on the streets. 

The final section of this article explores how waste pickers, particularly female waste pickers, take agency to mitigate against these hazards. The article concludes by asking what may be learnt about informal work and occupational hazards that holds relevance for the integration of waste pickers into municipal waste systems. Integrating waste pickers into municipal systems in just and inclusive ways, as the National Waste Picker Integration Guidelines [10] aim to do, requires the recognition of providing social protections for informal workers, including protections related to health. If waste pickers are to be integrated in ways that reduce their current occupational hazards, then waste pickers on the streets need access to proper sanitation, protective clothing, secure spaces to work, and the ability to take leave from work for medical or maternity care.

## 2. The Broader Context of Waste Picking, Health, and Occupational Hazards

All around the world, there are millions of people who work at the lower end of the recycling value chain; collecting, sorting, and selling reusable and recyclable materials they find in streets, landfills, waterways, and open dumpsites across all cities and small towns [11,12,13]. Known as waste pickers (or waste reclaimers), these workers make significant, but under-valued, environmental and economic contributions. To start with, waste-pickers contribute to mitigating the harmful effects of climate change by diverting waste to landfills through their reuse and recycling activities, thus reducing the GHG emissions from landfills and transport to landfills [10]. The tonnage that is diverted from landfills and incinerators from their labour should not be underestimated. For example, a Brazilian study indicated that waste pickers recycled and recovered approximately 80% of cardboard and 92% of aluminium (cited in [14]). A study by Al-khatib in Cairo (Egypt), reported that waste pickers recovered almost 6000 tons per day of recyclable material from the municipal solid waste system [15]. Early studies in South Africa indicated that waste pickers recovered nearly 80% to 90% of recycled materials from the country’s post-consumer products [16]. Given the increase in unemployment in South Africa, and the increase in waste picking as a form of livelihood, this diverted tonnage is likely to have increased considerably. The economic contribution from freeing up space in landfills is equally important to emphasize. A study in 2014 estimated that South African waste pickers enabled municipalities to save up to ZAR 748.8 million in landfill air space [16]. It is also important to note that in many countries, waste pickers provide a waste management and recycling service free of charge. Yet, despite this important work, waste pickers are denied forms of protection linked to formal work [17]. As with the majority of informal workers, they are forced to cover all their own expenses, including the costs of equipment and tools, and carry all the risks associated with ill health, accidents, or having no work [18]. Waste pickers also face significant levels of stigma and harassment in the places they work [19,20]. Waste pickers continue to “perform the most labour-intensive and least rewarding first steps of recyclables extraction from mixed wastes” [21] and, despite their critical contributions, work on the fringes of formal urban waste management systems. 

In South Africa, recent policy shifts demonstrate a clear acknowledgement of the vital role that waste pickers play in the country’s waste economy. Following a global movement to make visible the work of, and fight for the rights of, waste pickers, there has been increasing recognition of waste pickers in South Africa. There has been a concerted and combined effort from organised waste-picker associations such as the South African Waste Picker Association (SAWPA) and the African Reclaimer Organisation (ARO), environmental justice organisations such as groundWork, and researchers [22] to reshape the policy landscape on waste pickers’ work. This has resulted in the Department of Forestry, Fisheries, and Environmental Affairs publishing the official Waste Picker Integration Guidelines [10] aimed at municipalities. What the waste picker integration guidelines emphasise is the important need to expand the recycling economy in a way that recognises the value, dignity, and extensive expertise held by waste pickers, thus improving their job security, working conditions, and incomes through participatory processes [10].

Waste pickers in South Africa, and elsewhere, offer substantial environmental and health benefits to all people through their (often free) labour. Their work directly contributes to environmental preservation through reducing GHGs and conserving natural resources via recycling and has a positive impact on public health due to climate mitigation and the reduction in waste burning [10]. The health and occupational hazards of waste pickers have recently received some research attention. Waste pickers are exposed to toxic and dangerous materials in their work, including faeces, medical waste, rotten food, and broken glass. In landfills, waste pickers are also exposed to high levels of air pollution that can cause serious health ailments. These risks are amplified in instances where waste pickers work without protective clothing. 

In an extensive multi-site case study, Schenck et. al [14] explored how the management of South African landfills impacted the lives and livelihoods of the waste pickers who worked there. The findings from nine South African landfills painted a bleak picture. Few sites had access to water for waste pickers, only four sites had ablution facilities, and only one had shade or some form of protection from the elements for waste workers. In addition, the interviewed waste pickers identified many more workplace challenges. At unmanaged sites with no access control, waste pickers (particularly women) were harassed by drug users and gangsters, who frequently stole their goods or money. One of the greatest risks identified was being knocked over by one of the trucks that dropped off waste at the sites. Many also suffered from chest complaints due to inhaling dust, smoke, and dangerous chemicals. Half of the interviewed waste pickers also reported relying on food scraps from the landfill sites, but when this food is rotten, waste pickers can fall ill [14]. 

Many studies on the health impacts of waste picking focus on waste pickers who operate in landfills and dumpsites [13,22,23,24,25]. These offer important insights into occupational hazards and how to mitigate them. However, waste pickers also operate across the city in both residential and business districts. Scholarly literature on the occupational hazards of street waste pickers is less common. This paper poses the questions: What are the specific occupational challenges faced by waste pickers operating in the inner city of Durban? What risks do they encounter, and how do they navigate the congested streets that are occupied by several other players and controlled by a municipality that criminalises them? This paper makes use of a qualitative methodology, specifically an ethnographic paradigm, to investigate the occupational risks faced by street waste pickers in the inner city of Durban. 

## 3. Methodology and Waste Picking in the Inner City

This study adopted a critical qualitative research methodology. According to Norman Denzin [26], a critical qualitative enquiry is one committed to exposing and critiquing the forms of inequality and discrimination that operate in daily life through qualitative data and analysis. Specifically, we used ethnography as a way of understanding both perceptions around waste in the inner city of Durban and waste pickers’ lived experiences. 

This research was part of the Warwick Zero Waste Project, which aims to build connectivity and capacity among informal worker movements, civil society, university academics, and city officials so as to create opportunities for co-design and learning with informal workers regarding mitigation strategies for climate change in relation to waste in Durban. This paper draws from the ethnographic component of the first year of this project. Researchers immersed themselves into the waste pickers’ recycling activities in the sites and spaces in which they worked from April to May 2021. This included spending time in specific locations that waste pickers had negotiated as safe places to work, as well as walking with waste pickers who collected waste across an area of the city. While in the field, researchers did not just observe the participants, they also engaged as collaborators in collecting, ferrying, and sorting recyclables with participants. Researchers generated detailed fieldnotes after each visit to the field. 

The sample from which the analysis for this paper was derived consisted of 8 waste pickers. The relatively small sample was justified given the qualitative nature of this study, where an “illustrative” rather than a “representative” sample was preferred [27]. The qualitative methodology selected for this research produced rich, in-depth data on the lives and working experiences of waste pickers—shedding light on the meaning these workers attach to their work and their work environments. Unlike quantitative studies that demand larger sample sizes to justify the subsequent generalizations made, qualitative studies seek to produce in-depth accounts of lived experience to draw attention to the nuances of context and individuality. That said, theoretical saturation is an important concept to consider when determining an appropriate sample size in qualitative research. Saturation is achieved at the point at which the data collected no longer offers any new findings or themes [28]. In the case of this paper, the in-depth research undertaken with the waste picker sample was sufficient to reach a saturation point in terms of understanding the risks, hazards, and mitigation strategies engaged in by these informal workers. 

The 8 waste pickers who participated in this research, 7 women and 1 man, were purposefully sampled by groundWork, a partner organisation of the Warwick Zero Waste Project. The groundWork organisation has built close relationships with waste pickers in Durban over the years and has a close working relationship with SAWPA. The waste pickers selected did not necessarily belong to this association (although two did), but were rather selected on the basis of their geographical location; working in the inner city and around the informal street markets of Warwick. Five out of the eight participants in this study resided outside Durban, where they could pay for more affordable rentals. This meant that they had to maintain an unhealthy cycle of waking up early, collecting and sorting waste for the majority of the day, and then travelling back home. GogoXulu, one of the participants, explained that for the past 20 years she has had to wake up at 4 a.m. so she can make it to work by 6 a.m. every day to collect waste [29]. These ethnographic fieldnotes provided descriptive explanations of the study area and of the interaction of people in the space [30], as well as offering insights into some of the occupational hazards and mitigation strategies deployed by the waste pickers in the study.

The waste pickers in this study, outlined in Table 1 below, have been made anonymous. Given the country’s history and pervasive racial and class inequalities, waste pickers in South Africa are almost exclusively black Africans and are typically economically and socially marginalised [10]. This was also the case for the 8 waste pickers in this study.

In South Africa, workers of all ages and both genders engage in waste picking, although, as this study will highlight, there are important gender distinctions. Globally, female waste pickers face particular challenges in their work, emanating from gender inequalities within the sector and in their respective societies [10]. The challenges faced by female waste pickers are intersectional, and the hardships they face intersect with other forms of oppression and discrimination based on race, caste, nationality, and religion. As such, not all female waste pickers face the same challenges; some are more burdened than others [31].

In South Africa, younger, stronger men tend to dominate residential waste-picking sites. They have the advantage of the age and strength to walk far distances, push trolleys on uneven terrain, and carry heavy loads [10,32,33,34]. These men often do not have the same domestic responsibilities as women so are also able to work longer hours, resulting in male waste pickers typically earning higher incomes [11,31]. Women are more often found on landfill sites, or, as in this study, they operate in commercial sites, where they have forged relationships with business owners. Working in these spaces, women are safer from sexual harassment and crime. 

All the waste pickers in this study referred to their livelihood as a form of work and expressed pride in how this had enabled them to provide for their families and send children to school. The female waste pickers, whilst still experiencing forms of harassment by authorities and stigmatisation from the public, were distinct in their own eyes from the younger, homeless men who pick waste from the streets of the city. Many of these young men use a form of heroin sold on the streets called Whoonga and sell whatever recyclables they can find to buy-back centres and mobile middle agents to ensure their daily supply. Waste picking is a favourable option for many poor individuals who are struggling with heroin use disorders, as it is a form of self-employment that provides regular, daily cash to purchase drugs [35]. While these young men, frequently called the derogatory term *amapara* (roughly translated as ‘parasite’) engage in a daily “hustle” for drugs, they, like many of the waste pickers in this study, form part of the “labouring poor” in the country—those who engage in low paid work that does not easily allow for social mobility [35]. As this article outlines in the following sections, some occupational tensions emerge between female waste pickers in the inner city of Durban and these young men. In this article, these young men are referred to as *abashana* (nephews), a more considerate term that has also emerged from the street.

## 4. Occupational Hazards in the Inner City

All informal workers suffer a lack of social protection linked to their work, which increases occupational hazards and the risks associated with these [14]. This section, however, focuses on the specific occupational hazards that waste pickers endure as they navigate difficult environmental and social working conditions during the process of collecting, sorting, storing, transporting, and selling recyclable materials. These hazards are both physical and socio-psychological. 

### 4.1. Physical Hazards

Physical injuries are a common occupational health risk for all waste pickers, including street waste pickers. All of the waste pickers participating in this study revealed how they had been injured at one point or another while carrying out their work. These injuries ranged in terms of magnitude, and often happened during the process of collecting and sorting waste. Some were cut by sharp objects during the process of searching for waste in public bins, while others suffered serious physical strain. 

One of the waste pickers who participated in this research, Thandiwe, explained that she could no longer count the number of times she endured cuts from glass, pins on cardboard, and other mixed objects while collecting waste, especially from bins [36]. Her story is one that many waste pickers identify with. Waste picking involves sorting through unseparated, mixed waste, and as such, waste pickers often come across sharp materials. Waste pickers also work with plastic material, which requires them to use razor blades for the removal of sticky paper, barcodes, and price tags. Razor blade cuts are a common occurrence as a result [29]. Whilst these injuries may appear small initially, they can become infected and require medical treatment, as well as placing waste pickers at risk of tetanus. Waste pickers themselves are very aware of these risks. One of the women told us that her most difficult memory was of waste picking while pregnant. She felt a daily struggle to continue working to fend for herself and her other children [29] yet was also trying to ensure the safety of the child she was carrying in utero. Given that she had to save enough funds to sustain herself and her family during the birth and the postnatal recovery period, she had to increase her workload rather than take days of rest and recovery. She said this period of her work made her anxious and depressed [29]. These unimaginable stresses linked to physical endurance and the responsibility of caring for children are not uncommon in this occupation. Studies in Johannesburg and elsewhere have highlighted that female waste pickers are “more likely to report common mental health disorders than their male counterparts” [37]. 

Physical injuries also occur from the daily transport of heavy materials across long distances, particularly for the older women in this study. In the case of MamDlamini, the collecting, carrying, and ferrying of cardboard waste across the streets of Durban on her head caused serious health implications. She has an existing head injury caused from carrying bundles of waste long distances, which now negatively impacts on her work [38]. MamDlamini and a considerable number of other female waste pickers had to resort to using trolleys/carts as an easy-to-use mode of transporting heavy recyclable materials. However, this too is not without its risks. According to MamDlamini, a heavy loaded cart cannot be pushed with ease on the congested and broken pavements. Physical sprains also occur through the exertion required to push the trolley forward on uneven and broken pavements [38]. Moreover, waste pickers must negotiate extensive human traffic on pavements. To avoid the tensions that arise from trying to maneuver trolleys through busy sidewalks, many waste pickers prefer to push their trolleys on main roads in the city, which are smoother but dangerous because they are monopolised by vehicles, which have scant regard for pedestrians [38]. 

Considering that the majority of the participants in this study were elderly women, and that they had to work on a daily basis, many were at severe risk of long-term muscular-skeletal disorders, such as back pain, neck pain, and disorders affecting the arms and shoulders. These disorders from lifting heavy materials plague informal waste pickers worldwide [39]. Despite the hard-working conditions and hazards linked to this, waste pickers in a number of other studies reported that they preferred this occupation to other types of low-earning occupations (such as domestic work) because it afforded them the freedom to be their own boss and schedule their work around other priorities, such as childcare, and ensured that they were not subjected to potential racial abuse and exploitation from employers [10,14].

Against the backdrop of these risks, however, waste pickers have low health-seeking behaviour [25,40]. All informal workers suffer from a ‘no work no pay’ situation, and this can make waste pickers reluctant to seek medical attention for work-related injuries. According to the participants in this Durban study, a day away from work meant a reduction in income or no income at all. Waste pickers are therefore forced to work on daily basis. Being unable to take time off means no access to medical services and further exposure to the risk of infection, as well as exacerbating muscular or skeletal disorders. Over time, this results in accumulated health vulnerability and unchecked physical ailments. 

### 4.2. Stigma and Socio-Psychological Hazards

Some of the psychological hazards of not taking time off work to attend to family life or medical needs have already been mentioned. As will be discussed further on, female waste pickers must also navigate their fears for their own safety, in relation to government officials, police, strangers on the street, and the *abashana* who steal their goods. What all the waste pickers in this study shared, including the one man who participated, were feelings of being socially stigmatized for the work they participated in. They had all experienced being perceived as ‘dirty’ or ‘filthy’ because of their work with waste. All the participants in this study had received verbal insults of this kind when going about their daily work in the city. For MamDlamini, this frequently happened when she was pushing her materials on a trolley on busy pedestrian pavements. According to her, “people are too stubborn to give way” [38], and when asked to move, she had been verbally insulted by pedestrians, street cleaners, and shop owners. The waste pickers in this study worried that these verbal insults might turn into physical attacks. Whilst none of the waste pickers had experienced this form of violence themselves, they all knew of cases that had happened. Indeed, they have cause for concern in these cases, as the research literature indicates very high levels of both stigma and harassment for waste pickers in the places that they work [18,19,20,21].

### 4.3. Competition and Safety on the Street

Waste pickers are also in constant competition with other waste pickers, particularly the young men referred to here as ‘*abashana*’. Competing for space and a market with *abashana* poses real physical security concerns, especially for female waste pickers. The waste pickers that formed part of this project indicated that it was often *abashana* who intruded into their workspaces and stole their materials. Collecting and storing waste on the street is not secure. Nomkhosi [41] explained that she had been playing a game of hide and seek with, in her words, ‘*amaphara’* for years: “*Bayayitshontsha icardboard yami labafana, bacabanga ukuthi ngizele ukuzosebenzela bona*!” (“These boys steal my cardboard; they think I am coming here to work for them!”). 

*Abashana*, however, are not the only threat to keeping hold of one’s materials on the street. Thandiwe explained that her major threat was the municipality itself: “*Abasiboni njengabantu labayana*” (“They do not see us as people, those ones”) [36]. She told us that, in her view, the municipality thought waste pickers should be cleaned off the streets. Treated as illegal and informal rather than thanked for providing a much-needed service, waste pickers in the inner city feel harassed and monitored by police and other officials. Thandiwe [36] recounted that on several occasions her waste had been confiscated by the municipality on the pretext that she was ‘making the city dirty’. These forms of discrimination and stigma result in substantial occupational hazards for many waste pickers.

### 4.4. Environmental Health and Sanitation Hazards

Street waste pickers in Durban operate in spaces that are inherently risky. The nature of the work often means working on streets, or backyards and storage areas of shops where waste is kept. All the waste pickers in this study have poor access to functional public sanitation facilities. According to the research findings, most public toilets that waste pickers in inner-city Durban largely rely on are in a deplorable or dysfunctional state. The toilets in the markets of Warwick, for example, are in a constant state of disorder, either because of pipe blockages, broken taps, or water scarcity. Thus, waste pickers need to resort to using open spaces to relieve themselves. 

Water scarcity in workspaces translates to minimal or no handwashing with or without soap. The health consequences of this for people who work with waste are severe. Sadly, this is a common occupational hazard for waste pickers in the Global South [42,43]. Participants indicated that they came across and touched all manner of waste, including human waste. MamSibiya [44], for instance, had to go through meat packaging plastics, which contained blood deposits. Her materials and work site were thus a breeding ground for flies and other pathogens, which made her and other waste pickers prone to infections. 

The lack of access to adequate sanitation is not the only environmental hazard that these waste pickers face. Interestingly, this study found that street waste pickers also run the risk of respiratory issues linked to their work. While respiratory risks are often reported among waste pickers who work in dumpsites where there is consistent burning of waste [22], it was unexpected to find this as a concern for street waste pickers. This complaint was made by the female waste pickers who worked in enclosed refuse areas at the back of retail centres. Most of these areas do not have proper ventilation. Researchers noted in their fieldnotes how most of the spaces were exceptionally hot and not ventilated, and that in one case, when collection was delayed, the smell of rot from the wet cardboard was overwhelming. Some waste pickers who work with cardboard wet their material so that it weighs more at the buy-back centre, in the hope of receiving more money. Daily exposure to enclosed and poorly ventilated spaces that are specifically used for managing waste exposes waste pickers to respiratory risks linked not to burning, as is the case in landfills, but to mould and damp, causing long-term chest and respiratory irritation. 

## 5. Mitigation Strategies

In all forms of waste picking, there are significant risks. There are various biological, chemical, physical, and mental health hazards that waste pickers face in their work. It is important to note, however, that waste pickers also hold agency. They are savvy navigators of these streets and the social relationships that populate them. Waste pickers draw on their own and others’ extensive experiences of working on the streets to try and mitigate some of the occupational hazards that arise from their work. 

### 5.1. Relationship Building

Some of the waste pickers in this study established relationships with organisations and NGOs. During the COVID-19 pandemic, these relationships helped in securing sites within business premises, providing protective clothing such as boots and gloves, and health training. One NGO, Asiye eTafuleni, also provided hand-drawn carts/trolleys for transporting waste more easily and letters of support, which waste pickers used to obtain new business from retail stores in the inner city. Waste pickers recognise that in some cases, establishing relationships with NGOs can also provide a space for advocacy regarding more inclusive laws and policies to improve the safety of waste pickers [44].

Interestingly, the participants of this study also indicated that they built relationships with some of the *abashana* in the city. As described early on, these young men are often seen both as competition and as dangerous. Most of the waste pickers worried about theft from *abashana*. However, some of these young men form part of the supportive network that waste pickers have built to get through their daily rounds and earn a livelihood. The research fieldnotes showed that waste pickers built relationships with these young men and other informal workers in the city, such as car guards and informal traders, who helped watch over materials when needed [41]. Some waste pickers even employed *abashana* to deliver heavier materials such as cans to buy-back centres, who then brought back the money for a small fee [29].

GogoXulu had a long-standing relationship with a young man named Sizwe, who salvaged food from bins in the area where she worked. As the fieldnotes showed, this relationship was mutually beneficial:

“GogoXulu has a mother-child relationship with one young man, Sizwe. GogoXulu collects cardboard while he collects vegetables like tomatoes and green pepper in the market refuse area which he then recycles for sale. This relationship has meant that when Sizwe is away selling his goods, GogoXulu keeps guard and when GogoXulu goes to collect her cardboard for sorting, Sizwe will be watching over the rest”.[29]

While relationships such as the one between GogoXulu and Sizwe take time and provide mutual security for work on the street, they are also at times precarious. Sadly, during this research, Sizwe, who smoked heroin daily, stole from GogoXulu. A hard life on the streets, in the context of the limited resources available to informal workers, can strain social networks. This means that for waste pickers, creating social networks is a continuous strategy of renegotiation and rebuilding, rather than a stable network that accumulates trust and depth over long periods of time. In this sense, waste pickers in the inner city of Durban, over and above carrying out hard physical labour, perform a lot of emotional and social labour on a daily basis to mitigate against physical, psychological, and safety hazards linked to their occupation.

### 5.2. Secure Spaces 

Another key mitigation strategy for many of female waste pickers is to secure a more permanent space in the city. This helps with both supply (as people know where to find them to drop off cardboard) and storage, and it reduces the need to move large quantities of waste around the city. All waste pickers require Material Recycling Facilities (MRFs) if they wish to scale up their work and have a safe space to store and bundle waste for collection by buyers. The city of Durban, one of South Africa’s largest metropolitans, has only one MRF that we are aware of in the inner city. As a result, waste pickers must negotiate more secure spaces on their own. Women waste pickers often build relationships with retail stores through both collecting their cardboard waste and offering to clean the retailer’s waste area for free. Providing this free cleaning service sometimes pays off through being invited to use the space for their work, as long as it remains well-looked-after. For one of the waste pickers in this study, Mam Sibiya, this mitigation strategy resulted in more secure formal employment:

“Mam Sibiya used to be a transit waste picker. She would move around asking for cardboard from shops and vendors in the CBD. Several years into her trade, good forces connived, and Mam Sibiya was invited by management of a large butchery to join them officially on a contractual basis. This meant she would still do the same work, that is collecting and preparing all the cardboard box from the butchery. As a contract employee, Mam Sibiya is paid weekly by the company, for however many hours she chooses to put in. The returns from the sales of cardboard boxes remain hers”.[44]

Other waste pickers who had managed to operate in retail centers with designated refuse areas also said it offered some security from theft. Retail areas have security personnel and cameras, which helps with security concerns. One waste picker recounted two incidents whereby people had attempted to steal her cardboard boxes within one of the refuse areas. In both cases, the security personnel responded promptly and apprehended the alleged thieves [45]. One of the researchers also noted that “before the work started, we had to report to the workshop security control room to seek permission to work with Zinhle” [45]. Where waste pickers have successfully established relationships with formal businesses, they gain some form of social protection associated with formal work. Of course, in many ways this protection remains tenuous, as it is linked to specific relationships with store managers and/or security guards, which can easily change over time.

## 6. Conclusions

Decreasing risks to waste pickers’ health requires a multi-pronged approach that includes educating not only waste pickers (who, as this study showed, are already well aware of their occupational risks) but also local governments, businesses, and residents about how to support and work with waste pickers in their area. With this study, we hoped to shed some light on the daily occupational hazards faced by waste pickers in the inner city of Durban. Since waste pickers are a socially diverse group, it is important to map and research these risks within their local contexts [46]. This ensures that some of the obvious physical and environmental risks are addressed, but importantly recognises the mental health and psychological risks that emerge for waste pickers in their work in the inner city; in this case, regarding the stress of being stigmatised, not being able to take time off work when needed, and fearing for their own safety and the safety of their materials.

Like all informal workers, waste pickers are afforded no social protections linked to their work. They alone carry the costs of any accidents (cuts from sorting waste, muscular-skeletal disorders from heavy loads), medical attention (to treat infections and address respiratory issues from their work), and tools and equipment (trolleys, razors, bundling, and PPE) needed to make their work less hazardous. Waste pickers in South Africa and globally have mobilised around their right to decent work, and a more consistent government approach to recognising the social, economic, and environmental value of waste pickers has emerged. 

What this ethnographic study shows is that waste pickers already carry out both physical and emotional labour to build mitigation strategies to deal with a wide range of occupational hazards. Recognising this, it is critical to engage with waste pickers as experts themselves, regarding not just how to integrate their work into municipal waste systems, but also how this formal integration can address their health and occupational risks in ways that are responsive to their contextual needs.

## Figures and Tables

**Table 1 ijerph-19-12986-t001:** Outline of research sample.

Participant Name (Changed)	Approx. Age	Gender	Work Experience on the Street	Recycling Materials Collected
MamDlamini	40–50s	F	Over 10 years	Cardboard and plastic
MamMnguni	60–70s	F	10 years	Cardboard, paper, and plastic
Mr Mlomo	50–60s	M	31 years	Cardboard, paper, cans, and plastic
Nomkhosi	30–40s	F	Started in 2008, 13 years	Cardboard, paper, cans, and plastic
Zinhle	20–30s	F	Did not say	Cardboard, paper, and plastic
MamSibiya	40–50s	F	7 years	Cardboard, paper, cans, and plastic
GogoXulu	80s	F	Over 20 years	Cardboard and paper
Thandiwe	30–40s	F	8 years	Carboard and cans

## Data Availability

Not applicable.

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
