# Peer review of "Working with Waste: Hazards and Mitigation Strategies Used by Waste Pickers in the Inner City of Durban"

_ijerph, 2022, doi:10.3390/ijerph192012986_

Round 1

Reviewer 1 Report

Dear Author,

Please take into account:

Line 176: Please, use the number 8, not the word;

Line 178: Please, use capital letters for the beginning of the sentence, like G for the groundwork;

Line 195: number and name the table above it; complete the rows 1, 3 and 6 (from the Table) with s; erase commas from cans, and plastic; and the names of participants are better should be simplified using their initials;

Line 453: avoid personal expression in scientific articles, such as: we did, we address ..., use impersonal expression like it was done, it was realized. Line 463, same problem.

Line 466- 467: “What this ethnographic study shows is that waste pickers already do both physical and emotional labour to build mitigation strategies to deal with a wide range of occupational hazards” I think is better to be reformulate because the waste pickers are looking for solutions, methods, ways to make their work less risky, for their safety, it's a survival thing. It’s not something they proposed as development, progress of their community. I don't think we can talk about mitigation strategies, it's too much said.

Replace the names of the participants in the study with initials, to give the paper an impersonal, scientific character (so that it does not resemble a story).

Regards,

Reviewer 2 Report

The issue of waste pickers is extremely important, mainly because of the huge and underestimated role they play in society. Hence, it was reasonable for the authors to take up this topic.

The article, in my opinion, has many weaknesses. The sampling was purposive and the small number of participants in the activity does not allow to draw valuable scientific conclusions. The use of critical qualitative research methodology, would have had its justification if the research sample was more numerous. I understand that the study was extremely difficult, absorbing the authors, but even with their best intentions, I think the conclusions are not scientific. 

The study lacks a literature review organizing the knowledge gained on the characteristics of waste pickers and the division of hazards in this work. We do not know why such hazards were chosen and not others. In addition, the authors have not presented the work of other researchers in this area. 

The authors did not present the characteristics of the area in which the study was conducted, e.g., the population of Durban, the number of (approximate) waste pickers, their potential share of the waste collection market, etc. For the reader, the geographic area analyzed may be unfamiliar. 

Two additional points: 

1. sources should be scholarly (peer-reviewed), accessible to the reader - field notes have no such character (Missing Hunter 2020 - what is this source?).

2. the authors too often call local studies (bibliography items: 33, 35, 36) global and incorrectly generalize their observations to a broader scale than the studies conducted.

In my opinion, at this point, the article needs both expanded research and a thorough overhaul of the literature review, especially in terms of systematizing the previous work of other researchers. 
